# An Analytical Method for Elastic Modulus of the Sandwich BCC Lattice Structure Based on Assumption of Linear Distribution

**DOI:** 10.3390/ma16093315

**Published:** 2023-04-23

**Authors:** Jinqi Shang, Kangkang Wang, Dongyang Yan, Fengrui Liu, Linjuan Wang, Libin Zhao

**Affiliations:** 1School of Astronautics, Beihang University, Beijing 100191, China; 18710256089@163.com; 2Beijing Institute of Aerospace Systems Engineering, Beijing 100076, China; 3School of Mechanical Engineering, Hebei University of Technology, Tianjin 300401, China; 4Key Laboratory of Hebei Province on Scale-Span Intelligent Equipment Technology, Hebei University of Technology, Tianjin 300401, China

**Keywords:** sandwich BCC lattice, elastic modulus, analytical method, linear modulus distribution

## Abstract

An analytical method to predict the elastic modulus of the sandwich body-centered cubic (BCC) lattice structure is presented on the basis of the assumption of a linearly changing elastic modulus. In the constrained region, the maximum of elastic modulus used the elastic moduli of the BCC lattice element with plate constraints and is calculated with Timoshenko beam theory, the minimum used without plate constraints. In the rest of the constrained region, a linear function along the thickness direction is proposed to calculate elastic modulus. The elastic modulus of the unconstrained region is constant and it is the same as the minimum of the constrained region. The elastic modulus of the whole sandwich BCC lattice structure can be calculated theoretically with the elastic modulus of the constrained and unconstrained regions and a single-layer slice integration method. Six kinds of sandwich BCC lattice structures with different geometric parameters are designed and made by resin 3D printing technology, and the elastic moduli are measured. By comparing the predictions of the elastic modulus using the proposed analytical method and existing method with experimental results, the errors between the results of the existing method and the experimental results varied from 10.3% to 24.7%, and the errors between the results of the proposed method and the experimental results varied from 1.6% to 7.4%, proving that the proposed method is more accurate than the existing methods.

## 1. Introduction

Lattice structures, such as 3D-Kagome [1,2,3,4], eight-truss grid structure [5,6], body-centered cubic (BCC) [7], body-centered cubic with *Z*-direction struts (BCCZ) [8], simple cubic (SC) [9] and face-centered cubic (FCC) lattices [10], have large strength to mass ratios, good energy absorption and highly efficient energy storage [11]. The development of 3D printing technology has greatly enriched the types of lattice structures that can be manufactured. It has contributed to the application of lattice structures with tremendous potential in aerospace, medical equipment and other fields. The BCC lattice structure has the advantages of more reliable manufacturing, a simpler failure mode [12] than other lattices and has a broader application prospect. The BCC lattice structure is important to the designing and optimizing of structures, and so the mechanical properties of the BCC lattice structure need to be studied.

Many researchers studied the BCC lattice structures by conducting experiments. Dong [13] fabricated a BCC lattice structure and performed compression experiments. Liu et al. [14] designed a multilayer BCC lattice structure and experimentally validated that the multilayer BCC lattice structure had good compressive resistance. Tancogne-Dejean and Mohr [15] made a BCC lattice structure by laser melting 316L stainless steel and performed static and dynamic compression tests. The results showed that thinning the beam section could effectively enhance the energy absorption performance of the BCC lattice structure. Numerous researchers measured the mechanical properties of lattice structures through experiments [16,17,18,19]. Ling et al. [20] used 3D printing technology to make lattice structures of three different densities with two different polymer resins. They investigated its mechanical behavior through quasistatic and dynamic compression experiments, and the results showed that the responses of the two lattice structures predicted numerically were in good agreement with the experimental results. BCC lattice structure which belongs to porous structures has good dynamic performance [21]. If it is applied to the material structure design of military weapons [22], the mechanical effects, such as vibration, will be improved.

Many researchers also studied BCC lattice structures by simulations. Marschall et al. [23] and Deshpande et al. [24] built an FEM model and analyzed the static mechanism of the BCC lattice, and the predicted stiffness values were in good agreement with the test results. Mahbod et al. [25] proposed the FEM method of explicit nonlinear dynamic analysis and conducted a compression test on the BCC lattice structure, which verified the accuracy of the numerical prediction.

The mechanical analysis with experiments or FEM meant higher costs. Ptochos and Labeas [26] proposed an analytical formula for the elastic modulus based on Euler–Bernoulli and Timoshenko beam theories. The analytical method effectively reduced the solution time compared with FEM methods. Gümrük and Mines [27] predicted the initial stiffness of the BCC lattice structure by theoretical methods and claimed that the distribution of the elastic modulus was affected by the boundary conditions. The above are all studies on the mechanical properties of lattice. In actual structures, there are usually plate constraints [28,29], and the mechanical properties of sandwich BCC lattice structures are more practical. Gümrükd et al. [30] divided the sandwich BCC lattice structure into two regions to analyze mechanical properties. Zhang et al. [31] studied the shear modulus based on the equivalent homogenization methods of sandwich BCC lattice structures by using analytical, experimental and finite element methods. Yang et al. [32] assume two regions have two constant elastic moduli, respectively, and proposed an analytical method to predict the elastic modulus of the whole sandwich BCC lattice structure. However, the method cannot fully reveal the change law of the elastic modulus of the sandwich BCC lattice structure.

In this article, the assumption that the elastic modulus changes linearly is proposed to develop an analytical method for predicting the elastic modulus of the sandwich BCC lattice structure. The elastic moduli of six sandwich BCC lattice structures are calculated, and corresponding experiments are conducted to validate the analytical method.

## 2. Problem Description

The six sides of the BCC lattice structure are of an unconstrained boundary condition, as shown in Figure 1a. The structure consists of only struts. The sandwich BCC lattice structure is formed by adding plates to the upper and lower sides of the lattice structure, as shown in Figure 1b. The plates have constraining effects on the upper and lower sides of the lattice structure, and the other four sides of the lattice structure have an unconstrained boundary condition.

The mechanical properties of the BCC lattice structure with unconstrained boundaries are uniform, so the deformation along the *X*-direction is uniform under a means distributed single-axis load perpendicular to the upper and lower panels (as shown in Figure 1c). The deformation of the sandwich BCC lattice structure with upper and lower plates is different because plates affect the deformation of the lattice structure. The displacement contour is uneven along the *X*-direction, as shown in Figure 1d. Figure 1c,d are the displacement contours of panel ABCD and panel $A^{\prime }B^{\prime }C^{\prime }D^{\prime }$, as shown in Figure 1a,b.

Gümrükd et al. [30] divided the sandwich BCC lattice structure with diagonal struts into a constrained region (green region in Figure 2) and an unconstrained region (blank regions in Figure 2). The deformations of the struts in the two kinds of regions are different, as shown in Figure 1d. Yang et al. [32] believed that the deformation characteristics of different regions were different, that their elastic moduli were different and were two constants and consequently proposed an elastic modulus analysis model. The rationality of the constant elastic modulus in the two regions will be evaluated by an analysis of displacement characteristics in the next section.

## 3. Elastic Modulus of the Sandwich BCC Lattice Structure

### 3.1. Displacement Characteristic Analysis

The quadrilateral is marked $N_{1}N_{2}N_{3}N_{4}$ in Figure 3. The strain of the quadrilateral in the *X*-direction ($\varepsilon_{x}$) is recorded as $\sqrt{3}\left(u_{x1}-u_{x2}\right)/\left(2\sqrt{2}l\right)$, where $\left(u_{x1}-u_{x2}\right)$ is the difference in displacement between $N_{1}$ and $N_{2}$, the distance between $N_{1}$ and $N_{2}$ is $2\sqrt{2}l/\sqrt{3}$ and l is the length of the strut. The strain in the *Z*-direction ($\varepsilon_{z}$) of the quadrilateral is recorded as $\sqrt{3}\left(u_{z3}-u_{z4}\right)/\left(2\sqrt{2}l\right)$, where $\left(u_{x1}-u_{x2}\right)$ is the difference in displacement between $N_{3}$ and $N_{4}$, and the distance between $N_{3}$ and $N_{4}$ is $2l/\sqrt{3}$. The deformation coefficient η of the quadrilateral is defined as the ratio of the two strains $\varepsilon_{x}$ and $\varepsilon_{z}$. The coefficient can be calculated by $\eta =\left|\left(u_{x1}-u_{x2}\right)/\left(u_{z3}-u_{z4}\right)/\sqrt{2}\right|$. This deformation coefficient could describe the constraint degree of the plates on the lattice structure. This is because when the *X*-direction displacement of the quadrilateral near the plates is constrained, $\varepsilon_{z}$ is small, so the deformation coefficient of the quadrilateral is small; when the *X*-direction displacement of the quadrilateral is constrained less when away from the plates, $\varepsilon_{z}$ and the deformation coefficient of the quadrilateral are both larger.

After calculating all deformation coefficients of quadrilaterals, the contour of coefficient η of the diagonal panel $A^{\prime }B^{\prime }C^{\prime }D^{\prime }$ is drawn as shown in Figure 4. The deformation coefficient changes 0.7 (from 0.47 to 0.40) in the unconstrained region, which is very small. According to reference [32], this paper also uses the constant elastic modulus, which can be obtained by calculating the elastic modulus $E_{e}^{u}$ of the unconstrained element in Figure 5a. In the constrained region, the deformation coefficient changes 1.7 (from 0.40 to 0.23) and the deformation coefficients close to the plate is the minimum. With the increase in the distance to the plates, the deformation coefficient increases gradually along the thickness direction arrow, as shown in Figure 4. When approaching intersection *B* of the diagonal struts, the deformation coefficient is almost the same as that of the unconstrained region. Therefore, if the elastic modulus of the element close to the plate is assumed to be $E_{e}^{c}$, which can be obtained by calculating the elastic modulus of the constrained element in Figure 5b, the elastic modulus of point *B* is assumed to be $E_{e}^{u}$ and the modulus changes linearly. The modulus of the sandwich BCC lattice structure $E_{r}$ can then be expressed by the following linear variation formula:
(1)$$\left\{\begin{array}{l} E_{r}\left(z\right)=\frac{E_{e}^{u}-E_{e}^{c}}{b}z+E_{e}^{c}\;\;\;\;(\mathrm{constrained}\;\mathrm{region})\\ E_{r}\left(z\right)=E_{e}^{u}\;\;\;\;(\mathrm{unconstrained}\;\mathrm{region}) \end{array}\right.$$
where b is the distance between the point *B* and the boundary of the lattice structure, as shown in Figure 4.

### 3.2. Elastic Modulus of the Element under Two Kinds of Constraint Conditions

Next, the elastic modulus of the two elements in Figure 5 will be calculated. The cube size is a×a×a. The cross section of the strut is a circular surface of radius r and the length of the strut is l. The ratio r/l is the aspect ratio, which can be regarded as k.

For the element in Figure 5, the equivalent stress, strain and elastic modulus parameters are defined as follows:

The equivalent stress of element $\sigma_{z}^{e}$ can be expressed as:(2)$$\sigma_{z}^{e}=\frac{F_{z}}{S_{e}}$$
where $F_{z}$ is the *Z*-direction load of a single strut and $S_{e}$ is the cross-section of the element. The equivalent strain of element $\varepsilon_{z}^{e}$ can be expressed as:(3)$$\varepsilon_{z}^{e}=\frac{\delta_{z}^{e}}{h}$$
where $\delta_{z}^{e}$ is the *Z*-direction deformation of the strut under the load $F_{z}$ and h is the half height of the element (h=a/2).

The element equivalent elastic modulus in the *Z*-direction can be regarded as $E_{z}^{e}$ and the constitutive equation in *Z*-direction is:(4)$$E_{z}^{e}=\frac{\sigma_{z}^{e}}{\varepsilon_{z}^{e}}$$

The element elastic modulus $E_{z}^{e}$ depends on the elastic modulus of the constitutive material ($E_{m}$), Poisson’s ratio ($v_{m}$) and geometric size of the element strut (r,l). When calculating the modulus of the element in Figure 5, one of the struts will be analyzed because the structure is symmetrical.

Two coordinate systems are needed for calculating the element modulus. The global coordinate system consists of X-, Y- and Z-axes, which are used to analyze the equivalent properties of the element. The local coordinate system consists of $x^{\prime }$-, $y^{\prime }$- and $z^{\prime }$-axes and is used for analyzing the deformation of the strut. As shown in Figure 6, the $x^{\prime }$-axis is along the longitudinal direction of the strut. The $y^{\prime }$-axis in the green plane is perpendicular to the strut and the $z^{\prime }$-axis is perpendicular to the green plane. The green plane is perpendicular to the XY-plane and crosses through the intersection of the struts. The deformation analysis of the single strut is shown in Figure 7a,b.

A single strut without constraint plate is statically determinate. $F_{x^{\prime }}$ and $F_{y^{\prime }}$ can be written as functions of the load $F_{z}$ in Figure 6a. Then, the deformation and stiffness of the strut can be calculated in the $x^{\prime }y^{\prime }z^{\prime }$-coordinate system in Figure 7a. For the constrained element, the plate and strut together bear the external load $F_{z}$ and the strut becomes statically indeterminate. The calculations of $F_{x^{\prime }}$ and $F_{y^{\prime }}$ (in Figure 6b) need to introduce the deformation compatibility condition that the displacement of the strut along the horizontal plane is zero.

#### 3.2.1. Elastic Modulus of the BCC Element without Plate Constraint

The components of $F_{z}$ in the $x^{\prime }$- and $y^{\prime }$-directions at point A (as shown in Figure 6a) can be expressed as:(5)$$\left\{\begin{array}{l} F_{x^{\prime }}=F_{z}\cos\theta \\ F_{y^{\prime }}=F_{z}\sin\theta \end{array}\right.$$
where the angle between $F_{z}$ and the strut is θ:(6)$$\cos\theta =\frac{1}{\sqrt{3}},\sin\theta =\frac{\sqrt{2}}{\sqrt{3}}$$

In the local coordinate system $x^{\prime }y^{\prime }z^{\prime }$, the deformation $\delta_{x^{\prime }}^{e}$ and $\delta_{y^{\prime }}^{e}$ are shown in Figure 7a. The strut *OA* can be idealized as a cantilever beam with one fixed at point *O* and a free end at point *A* [32].

Under the influence of external load, the strut undergoes axial compression deformation and bending deformation. When the geometric size of the strut satisfies the condition of the short and thick beam, the shear deflection term should be introduced for a deformation analysis. The deformation $\delta_{x^{\prime }}^{e}$ and $\delta_{y^{\prime }}^{e}$ can be obtained based on the Timoshenko beam theory [32]:(7)$$\left\{\begin{array}{l} \delta_{x^{\prime }}^{e}=\frac{F_{x^{\prime }}l}{\pi r^{2}E_{m}}\\ \delta_{y^{\prime }}^{e}=\frac{F_{y^{\prime }}l^{3}}{12E_{m}I_{y^{\prime }}^{e}}+\frac{F_{y^{\prime }}l}{\kappa \pi r^{2}G_{m}} \end{array}\right.$$
where $I_{y^{\prime }}^{e}$ is the quadratic moment of inertia. $E_{m}$ and $G_{m}$ are the elastic modulus and shear modulus, respectively. κ is the shear coefficient of the Timoshenko beam. When the cross-section of the strut is a solid circular section, the shear coefficient of the beam is [32]:(8)$$\kappa =\frac{6\left(1+v_{m}\right)}{7+6v_{m}}$$
where $v_{m}$ is Poisson’s ratio of the constitutive material. The deformation along the direction of $y^{\prime }$ in Equation (7) can be written as [32]:(9)$$\delta_{y^{\prime }}^{e}=\frac{F_{y^{\prime }}l^{3}}{3E_{m}\pi r^{4}}+\frac{7+6v_{m}}{3}\frac{F_{y^{\prime }}l}{E_{m}\pi r^{2}}$$

As shown in Figure 7a, the deformation of the strut along the *Z*-direction can be expressed as:(10)$$\delta_{z}^{e}=\delta_{x^{\prime }}^{e}\cos\theta +\delta_{y^{\prime }}^{e}\sin\theta$$

Substituting Equations (2), (3) and (10) into Equation (4), the expression of the elastic modulus $E_{e}^{u}$ in the *Z*-direction of the element without constraints is as follows:(11)$$E_{e}^{u}=9\sqrt{3}\pi k^{4}\frac{E_{m}}{12k^{2}v_{m}+17k^{2}+2}$$
where k=r/l.

#### 3.2.2. Elastic Modulus of the BCC Element with Plate Constraint

The element with the plate is shown in Figure 5b and the equivalent stress, strain and elastic modulus are the same as those of the unconstrained elements in Equations (2)–(4).

Because of the effect of the plate constraint, $F_{x^{\prime }}$ and $F_{y^{\prime }}$ cannot be obtained by Equation (5). The plate cannot generate a *Z*-direction component force to support the *Z*-direction load, and only the strut can support it. The relationship of $F_{z}$, $F_{x^{\prime }}$ and $F_{y^{\prime }}$ (shown in Figure 6b) can be expressed as follows:(12)$$F_{z}=F_{x^{\prime }}\cos\theta +F_{y^{\prime }}\sin\theta$$

The plate is much stiffer than the strut, so the deformation $\left(\Delta s\right)$ of the plate in the XY-plane is almost 0. The analysis of $F_{x^{\prime }}$ and $F_{y^{\prime }}$ requires introducing additional deformation compatibility conditions as shown in Equation (13).
(13)$$\Delta s=\delta_{y^{\prime }}^{e}\cos\theta -\delta_{x^{\prime }}^{e}\sin\theta =0$$

Substituting Equations (7) and (9) into Equation (13), the relationship between the component forces $F_{x^{\prime }}$ and $F_{y^{\prime }}$ can be obtained.
(14)$$F_{x^{\prime }}=\frac{\sqrt{2}F_{y^{\prime }}\left(6r^{2}v_{m}+2r^{2}+l^{2}\right)}{6r^{2}}$$

According to Equations (12) and (14), the external load $F_{z}$ can be written as follows:(15)$$F_{z}=\frac{\sqrt{6}F_{y^{\prime }}\left(6r^{2}v_{m}+l^{2}+13r^{2}\right)}{18r^{2}}$$

The stress $\sigma_{z}^{e}$ of the constrained element can be obtained by substituting Equation (15) into Equation (2).
(16)$$\sigma_{z}^{e}=\frac{2\sqrt{6}F_{y\prime }\left(6r^{2}v_{m}+l^{2}+13r^{2}\right)}{9r^{2}a^{2}}$$

The deformation $\delta_{z}^{e}$ along the *Z*-direction can be obtained from Equation (10). The strain $\varepsilon_{z}^{e}$ of the constrained element can be obtained by substituting Equation (10) into Equation (3).
(17)$$\varepsilon_{z}^{e}=\frac{2(\delta_{x^{\prime }}^{e}\cos\theta +\delta_{y^{\prime }}^{e}\sin\theta )}{a}$$

By substituting Equation (7) and Equation (9) into Equation (17), the relationship between $\varepsilon_{z}^{e}$ and $F_{y^{\prime }}$ can be obtained.
(18)$$\begin{array}{l} \varepsilon_{z}^{e}=\frac{1}{3\pi ar^{4}E_{m}}(2l^{3}F_{y^{\prime }}\sin\theta +\sqrt{2}l^{3}F_{y^{\prime }}\cos\theta +14lr^{2}F_{y^{\prime }}\sin\theta \\ \;\;\;\;\;\;+12lr^{2}F_{y^{\prime }}v_{m}\sin\theta +2\sqrt{2}lr^{2}F_{y^{\prime }}\cos\theta +6\sqrt{2}lr^{2}F_{y^{\prime }}v_{m}\cos\theta ) \end{array}$$

By substituting Equations (18) and (16) into Equation (4), the expression for the elastic modulus $E_{e}^{c}$ can be determined as follows:(19)$$E_{e}^{c}=\sqrt{3}\pi k^{2}E_{m}\frac{6k^{2}v_{m}+13k^{2}+1}{18k^{2}v_{m}+21k^{2}+3}$$

### 3.3. Elastic Modulus of the Sandwich BCC Lattice Structure

Next, the elastic modulus of two sandwich BCC lattice structures in Figure 8 was calculated. The length, width and height of the sandwich BCC lattice structure were B,C and H. When the values of these three parameters are the same (H=B=C), the sandwich BCC lattice structure is unique in that eight diagonal struts intersect at the center point, as shown in Figure 8a. When these parameters satisfy H>B or H>C, the two diagonal struts intersect at a point, and eight struts have four intersections. Therefore, the constrained deformation region is a prism, as shown in Figure 8b. The equivalent elastic modulus of the sandwich BCC lattice structure was obtained by integrating Equation (1) along the thickness direction. To realize the calculation, it was necessary to introduce the assumption that the displacement in the *Z*-direction was uniform in the structure with the same *z*-coordinate.

#### 3.3.1. Elastic Modulus of the Sandwich BCC Lattice Structure (H=B=C)

Based on the assumption, the lattice structure can be cut into pieces, as shown in Figure 9a. Each piece included two parts, the constrained region and the unconstrained region, as shown in Figure 9b. Under the assumption of uniform displacement, the elastic modulus of a piece was calculated as follows:(20)$$E_{dz}\left(z\right)=\frac{E_{e}^{u}S_{cp1}+E_{r}\left(z\right)S_{cp2}}{S}$$

$S=S_{cp1}+S_{cp2}$ is the area of the single-layer slice. $S_{cp2}=4{\left(b-z\right)}^{2}$ is the area of the constrained region in the slice and $S_{cp1}=4b^{2}-4{\left(b-z\right)}^{2}$ is the area of the unconstrained region in the slice.

The equivalent elastic modulus $E_{z}^{s}$ in the Z-direction of the sandwich BCC lattice structure could be obtained by integral calculation $E_{dz}\left(z\right)$ as:(21)$$E_{z}^{s}=\frac{H}{2\int_{0}^{\frac{H}{2}}\frac{dz}{E_{dz}\left(z\right)}}$$

#### 3.3.2. Elastic Modulus of the Sandwich BCC Lattice Structure (H>B
or H>C)

As shown in Figure 8b, the lattice structure could be divided into three regions: the top region with height $H_{T}$, the middle region with height $H_{M}$ and the bottom region with height $H_{B}$. The equivalent elastic modulus of the three regions were $E_{T}$, $E_{M}$ and $E_{B}$. $E_{T}$ was equal to $E_{B}$ because the BCC lattice structure was symmetrical. The calculation method of $E_{T}$ was the same as that of $E_{z}^{s}$ in Equation (21). The equivalent elastic modulus $E_{z}^{s}$ can be expressed as follows:(22)$$E_{z}^{s}=\frac{H}{\frac{2H_{T}}{E_{T}}+\frac{H_{M}}{E_{M}}}$$

## 4. Validation of the Analytical Method

### 4.1. Experiment

(1)Specimens

According to the size relationship (H=B=C), a sandwich BCC lattice structure of 8 × 8 × 8 elements was designed to validate the analytical method in this research. Each element was a cube of 5 × 5 × 5 mm^3^ when the size of the lattice structure was 40 × 40 × 40 mm^3^. According to the size relationship (H>C), a sandwich BCC lattice structure of 12 × 8 × 12 elements was designed. Each element was a cube of 5 × 5 × 5 mm^3^ when the size of the lattice structure was 60 × 40 × 60 mm^3^. Six groups of resin specimens with different aspect ratios were made by 3D printing, as shown in Table 1. The thickness of the upper and lower plates of the sandwich BCC lattice was 3 mm. The resin used for 3D printing was Future 8200; its properties are shown in Table 2. The physical picture of the structures is presented in Figure 10.

(2)Experiment

The uniaxial compression experiment was conducted with a MARK-10 ESM303 electric loading frame. The displacement and the load were automatically recorded. The deformation of the lattice specimen during the experiment is shown in Figure 11.

As shown in Figure 11a, the elements along the diagonal lines (dashed lines) underwent obvious deformation. The deformation interface was consistent with the interface between the constrained region and the unconstrained region. As shown in Figure 11b, the transverse deformation of the upper and lower sides was almost 0 due to the constraint of the upper and lower plates, while the transverse deformation in the middle was the largest.

In the progress of experiments, the damage occurs along the interface between the constrained region and the unconstrained region, which means that there is a clear dividing interface caused by the two regions’ stress and strain. The damage mode is important to the design of the sandwich BCC lattice structures. This damage could be stopped by adding other types of lattice cells to reinforce the interface.

### 4.2. Result Comparison

The elastic modulus of the experimental specimens in Table 1 were calculated by the analytical method proposed in this research and the existing analytical method in ref. [32]. The two elastic moduli were compared with the experimental results, as shown in Figure 12 and Figure 13. The results of the proposed analytical method were in good agreement with the experimental results. However, there was a noticeable difference between the results of the existing analytical method in ref. [32] and the experimental results.

As shown in Figure 12 and Figure 13, the analytical results in ref. [32] are seemingly higher than the experimental results because the analytical method in ref. [32] only regarded the modulus of the constrained region as a constant, while the constant modulus is apparently higher than the average modulus based on the linear distribution method. In reality, the modulus of the unconstrained region is not constant, so the analytical method in this paper is still an approximation method. However, the results demonstrate that the approximation does not affect the accuracy.

The results of the comparison between the two analytical results and the experimental results are shown in Table 3. Table 3 demonstrates that the errors between the results of the analytical method in ref. [32] and the experimental results varied from 10.3% to 24.7%. The analytical results from the proposed method were in close agreement with the experimental results, with errors ranging from 1.6% to 7.4%. The accuracy of the method developed in this paper was evidently better. The results illustrated that the assumption of a linear distribution of the elastic modulus was more applicable to these two kinds of lattices.

The analytical method proposed in this paper improves the efficiency and accuracy of predicting an elastic modulus. It has great implications for designing the sandwich BCC lattice structures. The analytical method could become the basis of the subsequent uncertainty analysis which can help engineers better design and optimize the lattice structure and improve the performance and safety of the structure. The analytical method can also be developed to predict the elastic modulus of other periodic multi-bar lattice structures.

## 5. Conclusions

In this paper, the linear distribution of the elastic modulus was proposed as an assumption for the sandwich BCC lattice structures. An analytical method was established for predicting the elastic modulus of a sandwich BCC lattice structure with the constraint of a plate. After designing six kinds of sandwich lattice structures with different aspect ratios, the moduli were calculated and measured via a uniaxial compression experiment. The results demonstrated that, compared with the analytical results based on the theory of a uniform modulus distribution, the analytical results based on the assumption of a modulus with a linear distribution were in better agreement with the experimental results. Consequently, the correctness of the proposed analytical method was validated. Therefore, the highlights of the present work can be summarized as follows:(1)The assumption of a linearly changing elastic modulus is proposed for predicting the elastic modulus of the sandwich BCC lattice.(2)The elastic moduli of the elements of a BCC lattice structure with and without plate constraints are analyzed.(3)The elastic modulus of the sandwich BCC lattice is calculated with single-layer slice and integration methods.

In actuality, the analytical method is not only suitable for the sandwich BCC lattice structure, but also suitable for other periodic struts lattice structures. By considering different conditions, the periodic structs lattice structures can be analyzed by dividing different regions. The analytical method may become the basis of a subsequent uncertainty analysis. It will be beneficial for designing and optimizing the lattice structure and improve the performance and safety of the structure.

## Figures and Tables

**Figure 1 materials-16-03315-f001:**
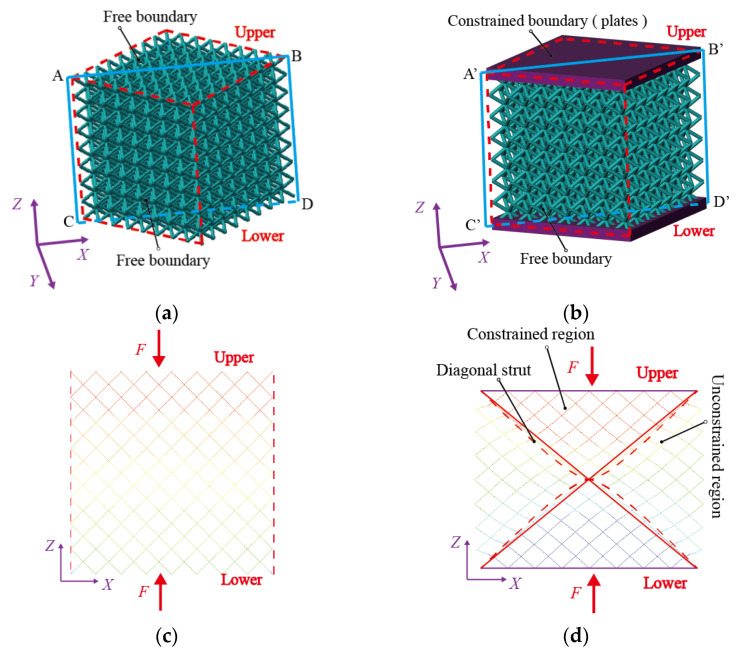
The BCC lattice structure model and mechanical properties: (**a**) the BCC lattice structure, (**b**) the sandwich BCC lattice structure, (**c**) displacement contour of the BCC lattice structure and (**d**) displacement contour of the sandwich BCC lattice structure.

**Figure 2 materials-16-03315-f002:**
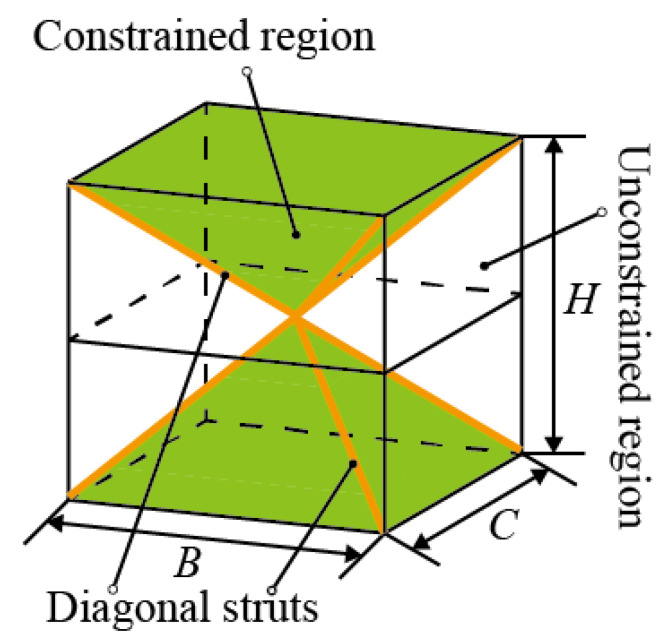
Two regions of the sandwich BCC lattice structure.

**Figure 3 materials-16-03315-f003:**
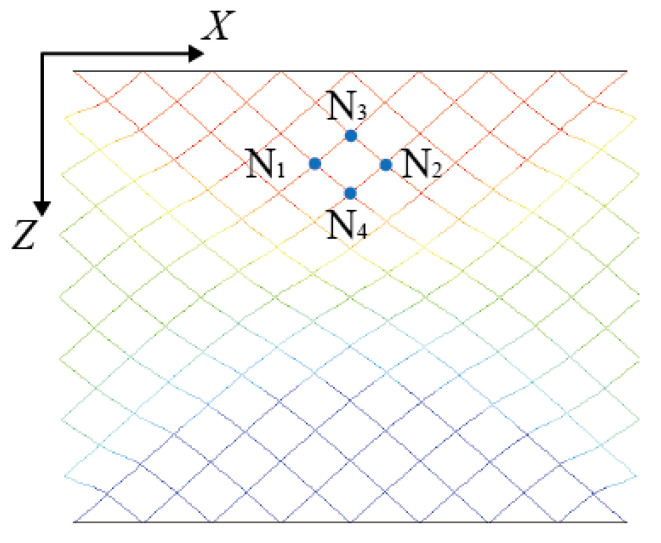
Schematic diagram of the four points of the quadrilateral.

**Figure 4 materials-16-03315-f004:**
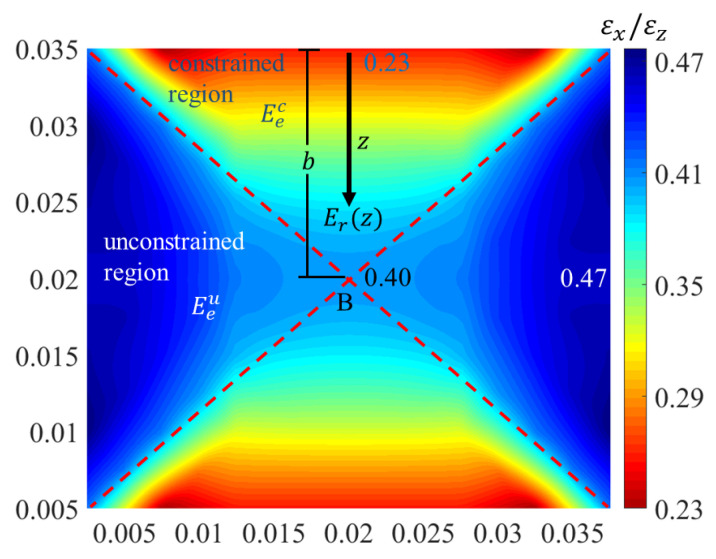
Deformation coefficient contour of the lattice with plate constraints.

**Figure 5 materials-16-03315-f005:**
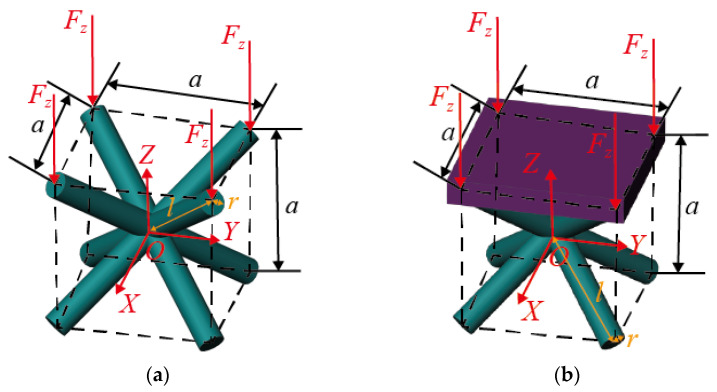
Two kinds of elements: (**a**) unconstrained BCC element, (**b**) constrained BCC element.

**Figure 6 materials-16-03315-f006:**
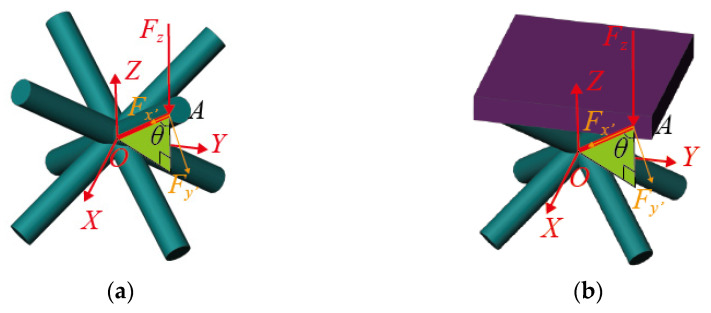
Two coordinate systems of elements: (**a**) unconstrained BCC element, (**b**) constrained BCC element.

**Figure 7 materials-16-03315-f007:**
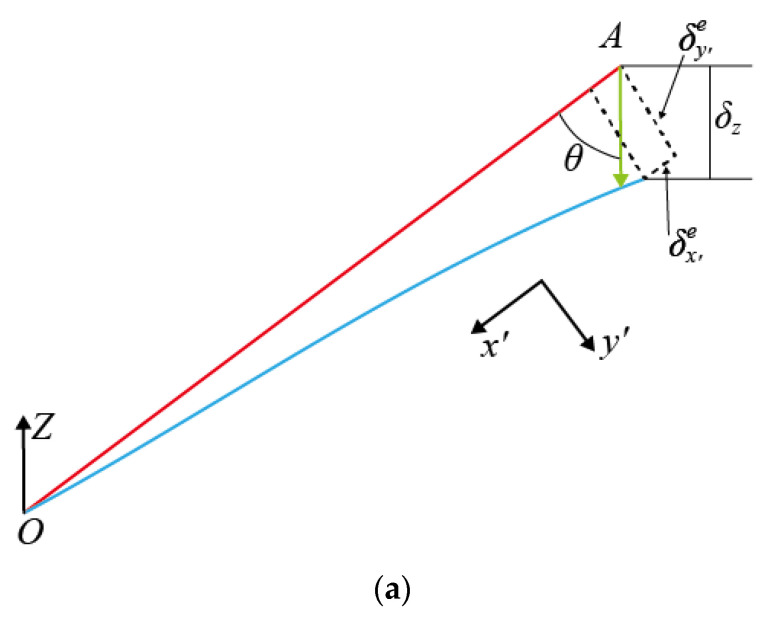
Deformation of two kinds of struts: (**a**) deformation of the strut of the unconstrained BCC element, (**b**) deformation of the strut of the constrained BCC element.

**Figure 8 materials-16-03315-f008:**
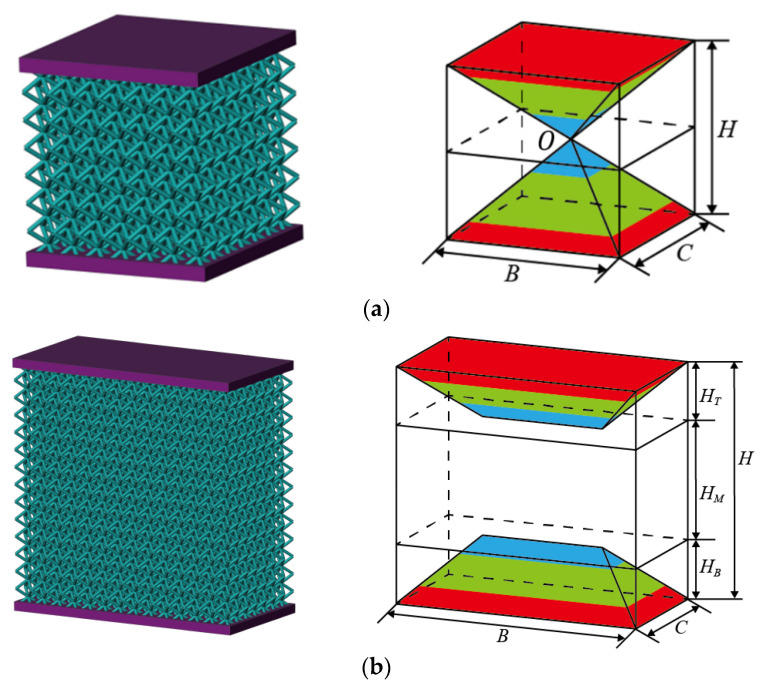
Models and sizes of different kinds of sandwich BCC lattice structures: (**a**) H=B=C, (**b**) H>C or  H>B.

**Figure 9 materials-16-03315-f009:**
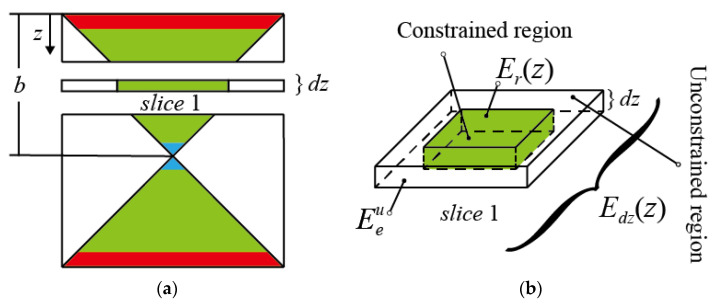
Slice analysis of the cubic sandwich BCC lattice: (**a**) slices of the cubic sandwich BCC lattice, (**b**) modulus distribution of a single-layer slice.

**Figure 10 materials-16-03315-f010:**
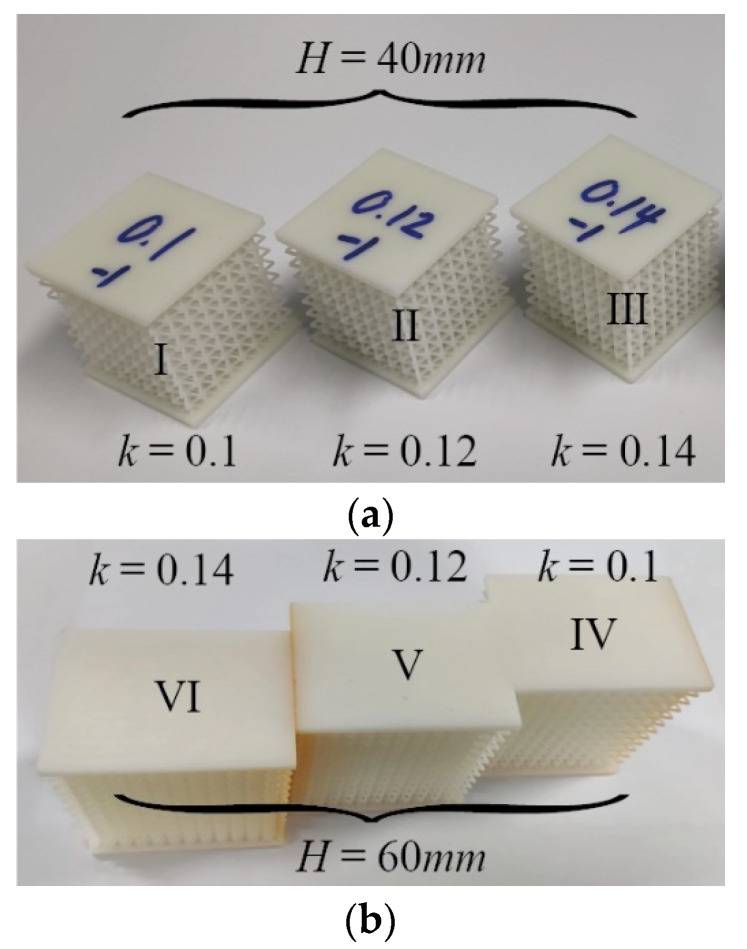
3D-printed specimens of the sandwich BCC lattice structures: (**a**) B=C=H, (**b**) H>C=B.

**Figure 11 materials-16-03315-f011:**
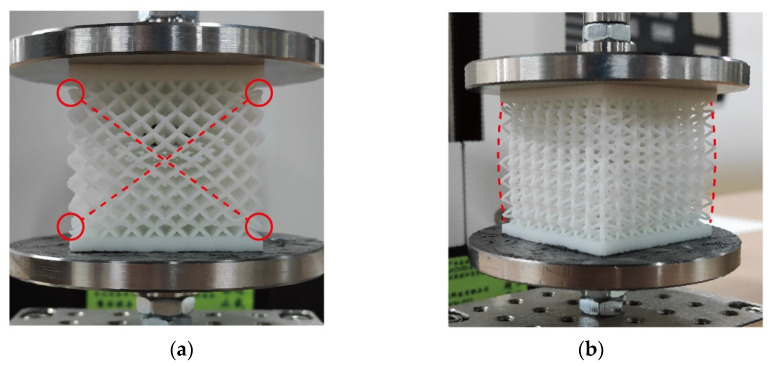
Deformation of the sandwich BCC lattice during the experiment: (**a**) location of failure in the lattice and (**b**) deformation of the lattice.

**Figure 12 materials-16-03315-f012:**
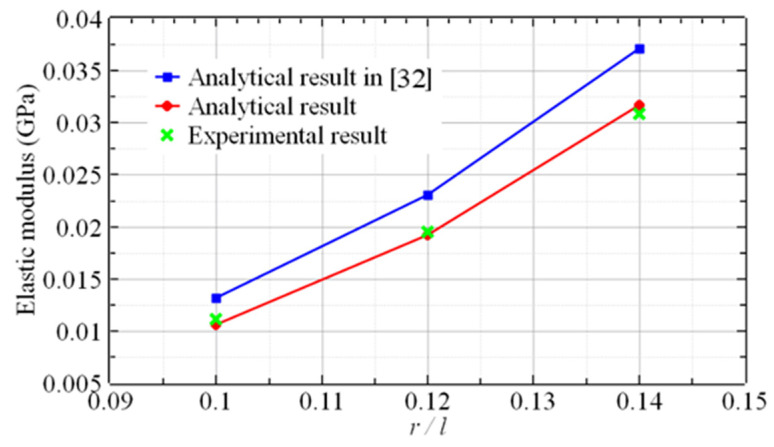
The elastic modulus vs. the aspect ratio of the sandwich BCC lattice (B=C=H).

**Figure 13 materials-16-03315-f013:**
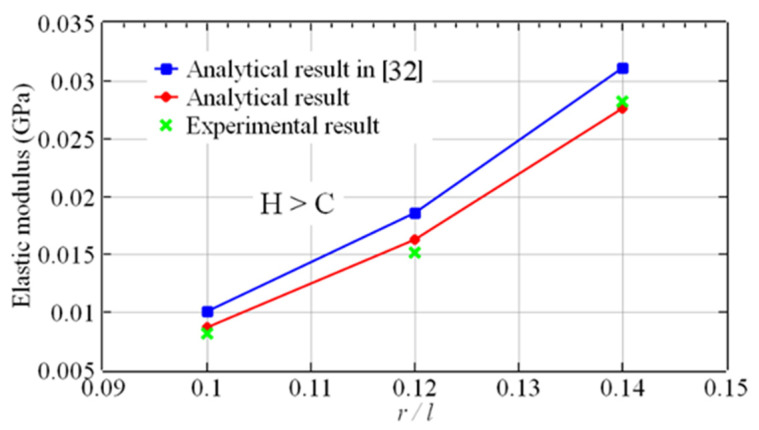
The elastic modulus vs. the aspect ratio of the sandwich BCC lattice (H>C=B).

**Table 1 materials-16-03315-t001:** Geometries of the printed BCC lattices.

No.	Element Number	Strut Diameter (mm)			Element Size (mm)	Strut Length (mm)	*k*
		Min.	Max.	Aver.			
I	8 × 8 × 8	0.88	0.97	0.91	4.97	4.30	0.10
II	8 × 8 × 8	1.03	1.07	1.06	4.98	4.31	0.12
III	8 × 8 × 8	1.20	1.24	1.23	4.97	4.30	0.14
IV	12 × 8 × 12	0.84	0.91	0.88	4.96	4.29	0.10
V	12 × 8 × 12	1.02	1.05	1.04	4.98	4.31	0.12
VI	12 × 8 × 12	1.22	1.24	1.23	4.97	4.30	0.14

**Table 2 materials-16-03315-t002:** The mechanical properties of material.

Parameter Name	Parameter Value
Tensile modulus/MPa (ASTM D 638)	2642
Poisson’s ratio (ASTM D 638)	0.42

**Table 3 materials-16-03315-t003:** Analytical and experimental results for the elastic modulus of the sandwich BCC lattice.

No.	k	Modulus from Experiments (MPa)	Method in This Research		Method in Ref. [32]	
			Modulus (MPa)	Error (%)	Modulus (MPa)	Error (%)
I	0.1	11.1	10.6	5.1	13.2	18.9
II	0.12	18.9	19.2	1.6	23.1	22.2
III	0.14	30.8	31.7	2.7	37.1	20.4
IV	0.1	8.1	8.7	7.4	10.1	24.7
V	0.12	15.2	16.3	7.2	18.6	22.3
VI	0.14	28.2	27.6	2.1	31.1	10.3

## Data Availability

All data are reported in the article.

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
