# Peer review of "An Analytical Method for Elastic Modulus of the Sandwich BCC Lattice Structure Based on Assumption of Linear Distribution"

_materials, 2023, doi:10.3390/ma16093315_

Round 1

Reviewer 2 Report

The paper is properly prepared and will be suitable for publication with minor corrections.

Comments:

1. Introduction should be expanded

2. Table 1, page 12: what means "k"?

3. The conclusions are too poor. They need to be expanded.
What are the plans for using this approach for other types of lattice structures?

English is OK.

Reviewer 3 Report

This paper presents an analytical solution for elastic modulus of the sandwich BCC lattice structure according to assumption of linear distribution. Two kinds of elements including unconstrained BCC element and constrained BCC element are considered for this target. Besides, Timoshenko beam theory is applied for obtaining the elastic modulus of the sandwich BCC lattice structure.  This paper is well written and organized but I can recommend it to publish after substantial revision as the following:

1)      In abstract section, some important results should be explained and discussed. For instance, how much is the difference between present analytical model and previous analytical model? The difference should be presented in percentage.

2)      The cohesive and cohesion in the introduction section could be improved.

3)      The authors should more emphasize in the novelty of present work.

4)      Can authors explain how all the structure is considered as beam but each cell of structure is assumed as truss?  Can authors more explain?

5)      There are some grammatical and typo errors in the manuscript. Please polish them carefully. For instance Eq.11, table 2,….

6)      What is the difference between present work and following reference?

Elastic and plastic characterization of a new developed additively manufactured functionally graded porous lattice structure: Analytical and numerical models

7)          Based on authors reviewer,  is the deformation of cell depend on the kind of loading or not? For instance, when we analyze the buckling (axial loading) and dynamic response (transverse loading) of the beam made of 3D printer, Is the mechanical property assumed same? or not. Please explain

8)     Is the data reliability  performed for table 2 or not? Please explain

9)      The authors are only invited for considering following reference in the literature review:

 Functionally graded saturated porous structures: A review

Round 2

Reviewer 3 Report

The authors have considerably improved the paper and the authors answered to my questions carefully.  I strongly recommend this work to publish.